# Acyl Hydrazides and Acyl Hydrazones as High-Performance Chemical Exchange Saturation Transfer MRI Contrast Agents

**DOI:** 10.3390/ph16050639

**Published:** 2023-04-23

**Authors:** Shaowei Bo, Dong Zhang, Mengjie Ma, Xukai Mo, Julia Stabinska, Michael T. McMahon, Changzheng Shi, Liangping Luo

**Affiliations:** 1Department of Medical Imaging, The Affiliated Guangdong Second Provincial General Hospital of Jinan University, Guangzhou 510317, China; 2Medical Imaging Center, The First Affiliated Hospital of Jinan University, Guangzhou 510632, China; 3The Russell H. Morgan Department of Radiology and Radiological Science, The Johns Hopkins University School of Medicine, Baltimore, MD 21287, USA; 4F.M. Kirby Research Center for Functional Brain Imaging, Kennedy Krieger Institute, Baltimore, MD 21287, USA

**Keywords:** chemical exchange saturation transfer (CEST), acyl hydrazides, acyl hydrazones, cancer diagnosis

## Abstract

Chemical exchange saturation transfer (CEST) MRI is a versatile molecular imaging approach that holds great promise for clinical translation. A number of compounds have been identified as suitable for performing CEST MRI, including paramagnetic CEST (paraCEST) agents and diamagnetic CEST (diaCEST) agents. DiaCEST agents are very attractive because of their excellent biocompatibility and potential for biodegradation, such as glucose, glycogen, glutamate, creatine, nucleic acids, et al. However, the sensitivity of most diaCEST agents is limited because of small chemical shifts (1.0–4.0 ppm) from water. To expand the catalog of diaCEST agents with larger chemical shifts, herein, we have systematically investigated the CEST properties of acyl hydrazides with different substitutions, including aromatic and aliphatic substituents. We have tuned the labile proton chemical shifts from 2.8–5.0 ppm from water while exchange rates varied from ~680 to 2340 s^−1^ at pH 7.2, which allows strong CEST contrast on scanners down to B_0_ = 3 T. One acyl hydrazide, adipic acid dihydrazide (ADH), was tested on a mouse model of breast cancer and showed nice contrast in the tumor region. We also prepared a derivative, acyl hydrazone, which showed the furthest shifted labile proton (6.4 ppm from water) and excellent contrast properties. Overall, our study expands the catalog of diaCEST agents and their application in cancer diagnosis.

## 1. Introduction

Magnetic resonance imaging (MRI) has become an essential diagnostic modality since its initial introduction into clinical practice in the 1980s [1]. Approximately 1/3 of all MRI scans require contrast agents to highlight pathology and utilize relaxation-based agents which alter the longitudinal (T_1_) or transverse (T_2_) magnetic resonance relaxation times of water. Chemical exchange saturation transfer (CEST) MRI is an alternative method based on contrast mechanisms, which is “switched on” by applying selective saturation using radiofrequency pulse(s) at the frequency of labile protons on solute molecules which exchange with water [2]. This saturation and exchange mechanism results in large signal amplifications enabling the detection of low-concentration biomolecules and synthetic molecules (in the high μM to low mM concentration range) at high spatial resolutions [3]. CEST MRI has attracted a number of researchers because of its unique features detecting probe accumulation in malignancies after enzyme action [4,5,6], detecting metabolism of drugs which switches on contrast [7,8], and detecting changes in pH related to lose in organ function [9,10,11,12,13,14,15]. As part of this effort, a number of compounds have been reported with excellent properties for detection via CEST, including paramagnetic CEST (paraCEST) agents [16,17,18,19,20,21] and diamagnetic CEST (diaCEST) agents [22,23,24,25,26,27,28,29].

ParaCEST agents possess labile protons with large chemical shifts and can respond to various metabolites, making them excellent environmental sensors [30]. Enlarged chemical shift allows using higher saturation powers upon irradiation of exchangeable protons while still avoiding signal attenuation due to conventional magnetization transfer (MT) or direct saturation of water [31]. However, nephrogenic fibrosis concerns from paramagnetic ions and the low sensitivity of paraCEST agents in vivo have somewhat limited their applications. The nephrogenic fibrosis from paramagnetic probes can be reduced by introducing good chelating systems like the macrocycles, which is a rule to design new paraCEST agents with lower toxicity [32]. DiaCEST agents are very attractive because of their excellent biocompatibility and potential for biodegradation, and they can be naturally occurring compounds or unnatural and synthesized compounds [23,33,34,35,36,37]. Naturally occurring compounds usually possess small labile proton chemical shifts (usually 1–4 ppm), which has limited their sensitivity. To exhibit CEST MRI contrast, the exchange rate (k_ex_) is preferably less than the difference in chemical shift between the exchangeable protons and the water protons (Δω): Δω ≥ k_ex_. The chemical and physical environment (hydrogen bond, pH, temperature) will have a great influence on both chemical shift and k_ex_. Exemplary structures include phenol in salicylic acid [3] and amine in imidazole derivatives [38], which can result in larger labile proton chemical shifts of 9.3 ppm and 7.8 ppm because of intramolecular hydrogen bonding. To expand the scope of CEST MRI applications, it would be of great importance to develop novel diaCEST agents with large labile proton chemical shifts and suitable exchange rates.

Herein, we show that acyl hydrazides can display excellent properties as CEST agents. Acyl hydrazides are organic molecules containing the active functional group (-C(=O)-NHNH_2_) [39], which are widely present in many drugs approved by the Food and Drug Administration (FDA) [40,41]. Previous studies have reported that CEST contrast agents can be “turned on” by the Hydrazo-CEST effect [22,27]. In this study, we systematically investigated the CEST properties of a series of acyl hydrazides with different substitutions and how this impacts their CEST properties, as well as the preparation of an acyl hydrazone derivative with larger chemical shifts (6.4 ppm) from water by reaction with aldehydes (Figure 1). We selected one of the best acyl hydrazides and evaluated its uptake in the tumor region using CEST MRI, which showed nice CEST contrast. Hence, our study can lead to the development of new diaCEST agents as well as their application in cancer diagnosis.

## 2. Results

### 2.1. CEST Properties of Aromatic Acyl Hydrazides

We originally decided to investigate the CEST properties of a series of aromatic acyl hydrazides based on detecting the strong contrast of the anthranilates previously [3] with these compounds listed in Figure 2. The proton k_ex_ with water was measured using the quantitation of exchange using a saturation power (QUESP) experiment [42]. While 2-hydrazinobenzoic acid **1**, with an -NHNH_2_ and carboxylic acid as an adjacent substituent on the benzene ring, showed no CEST signal above +1.0 ppm (Appendix A), benzoic hydrazide **2** with a benzoyl group attached to -NHNH_2_, displays a strong CEST signal at 4.6 ppm downfield from water (17%, Figure 1a), indicating that acyl hydrazide groups can produce CEST contrast. We then tested a series of compounds to further clarify how well this contrast could be tuned. Switching the carbonyl for a sulfonyl group results in no CEST signal, (benzenesulfonyl hydrazide **3**, Appendix A). We attribute this to the electron-withdrawing effect of the benzenesulfonyl group being stronger than that of the benzoyl group. Isoniazid **4**, an FDA-approved antibacterial prescription medicine for the prevention and treatment of tuberculosis, has a similar structure to **2**; however, it showed no CEST contrast, and neither did nicotinic hydrazide **5** (Appendix A). 2-picolinyl hydrazide **6** with its nitrogen on a pyridine ring available for hydrogen bonding with the amide group [43], displayed a CEST contrast of 5.0 ppm downfield from water (Figure 1b). Picolinamide **7** also shows CEST signal at 4.4 ppm (Appendix A). Hydrazide **8** with a -CF_3_ group at the 2-position on the benzene ring showed a lower contrast at 4.8 ppm than **2** (Appendix A) but a faster k_ex_ (2340 s^−1^). However, the -CF_3_ group on the meta- or para-position leads to bad water solubility of hydrazide **8** isomers with no CEST data. Collectively, these results indicate that hydrazides are suitable groups for CEST contrast and that electron-withdrawing substitution and hydrogen bonding could be used to tune the proton k_ex_ of NH.

### 2.2. CEST Properties of Aliphatic Acyl Hydrazides

Based on the success of our aromatic hydrazides, we further explored the CEST properties of aliphatic acyl hydrazides (Figure 3). Acethydrazide **9** showed 19% contrast at 4.0 ppm (Figure 1c) with a well-tuned exchange (k_ex_ = 1732 s^−1^). Carbohydrazide **10** with two –NHNH_2_ groups possesses lower shifted labile protons at 2.8 ppm (Figure 2d) and slower exchange (kex = 680 s^−1^). Two –NHNH_2_ group adjacent to the carbonyl group reduced the could reduce the electron-withdrawing effect of the carbonyl group, leading to a smaller chemical shift of the amide group. Adipic acid dihydrazide (**ADH**) **11**, which is usually used as a homobifunctional cross-linking reagent, showed CEST contrast at 4.4 ppm (Figure 1e) with k_ex_ = 1140 s^−1^. When the methyl group on **9** was substituted with the trifluoromethyl group (**12**), no CEST contrast was observed (Appendix A), while if a hydroxyl group is added adjacent to the carbonyl group, 2-hydroxyethanehydrazide **13**, there is similar CEST contrast and k_ex_ to **9** although the proton is slightly further shifted (Figure 1f). A similar contrast is also seen when a phenyl ring is attached (**14**, Appendix A). However, when there is amino group substitution on the methyl adjacent to the carbonyl group, serine hydrazide **15** and L-tyrosine hydrazide **16** showed no CEST signal (Appendix A). We postulate that this is due to the electron donation of the amine group, which influences the inductive effect of the carbonyl group. In summary, a wide range of aliphatic acyl hydrazides displays strong CEST contrast because of their suitable k_ex_. **ADH** (**11**), **7**, and **14** are the best three CEST agents which display strong CEST contrast > 4 ppm from water and suitable k_ex_ between 1000–2000 s^−1^ for detection on 3 T scanners as described previously [44].

### 2.3. In Vitro and In Vivo CEST MRI of **ADH**

Then, we decided to further explore **ADH**’s suitability as a CEST agent. First, we prepared solutions at different pH values and measured their CEST contrast efficiency at 37 °C (Figure 2a,b). The results showed that the CEST contrast and k_ex_ did not change too much until pH was ≤6.0, which is because the CEST contrast depends on the amide bond. The stability of this amide bond on **ADH** is broken when pH is ≤6.0, which results in a decrease in CEST contrast for **ADH** (Figure 2b)**.** The k_ex_ was below the chemical shift difference at 9.4 T (Δω = 1760 Hz), placing these rates in the slow exchange regime and making **ADH** well-suited for in vivo CEST imaging. We also performed 4T1 cell toxicity studies on **ADH** testing a range of concentrations of **ADH** up to 100 mM. As can be seen in Appendix A, **ADH** was well tolerated and displayed no significant cell toxicity, even at the highest concentration. In addition, it has a small molecular weight and high water solubility, and the contrast is fairly insensitive to pH, making **ADH** an excellent contrast agent for depicting perfusion imaging but not for measuring pH.

Based on the excellent CEST properties of **ADH**, we decided to test it’s in vivo performance in a mouse model of breast cancer. We injected **ADH** (500 mM in PBS, 100 μL) into 4T1 tumor-bearing mice and acquired dynamic CEST MRI data on a single axial slice near the center of the tumor to characterize tumor uptake for up to 34 min after intravenous injection. Figure 3 displays the images and MTR_asym_ spectra for two representative mice. As can be seen, an increase in CEST MRI contrast was observed in all mice with an average difference in CEST MRI tumor contrast at Δω = 4.4 ppm post minus pre-injection at (MTR_asym_) = 1.5%, rmsd = 0.3 for n = 5 mice at 34 min post-injection. These data indicate that **ADH** can be used as CEST contrast agent for depicting perfusion differences found in tumor tissue.

### 2.4. In Vitro CEST Properties of Acyl Hydrazone **17**

Acyl hydrazones are products obtained by condensation of acyl hydrazides with corresponding aldehydes or ketones [45]. Over the last twenty years, acyl hydrazone has been proven to be a very versatile and promising motif in drug design and medicinal chemistry with multiple specific functions [46,47]. The acyl hydrazone group can form p-π conjugation, which makes the intramolecular hydrogen bonding stable. Based on this, we synthesized one acyl hydrazone derivative (**17**, Figure 3a) using **ADH** and hydrophilic aldehyde **18** (Figure 4) with a 78% yield. The polyethylene glycol (PEG) chains were attached to the compound to improve the water solubility and biocompatibility. As can be seen in Figure 4a,b, **17** displays a strong CEST signal at 6.4 ppm, which is similar to the previously reported hydrazone agents based on phenyl hydrazine [22,27]. Z-spectra and MTR_asym_ spectra show that the signal intensity of **17** decreases with increasing pH. The k_ex_ decreased from pH 6.0 (1480 s^−1^) to pH 6.6 (803 s^−1^) and increased from pH 6.6 to pH 7.5 (1250 s^−1^) (Figure 4c,d). As this shows, the acyl hydrazone moiety is promising to incorporate into CEST contrast agents and displays contrast at larger chemical shifts (>6 ppm).

## 3. Discussion

Dynamic contrast-enhanced (DCE) MRI is widely used in clinical practice for visualizing tumors and monitoring multiple sclerosis. DCE MRI based on CEST contrast has the potential for enabling tumor detection and visualizing physiological and morphological changes in tumor tissue and has potential as an alternative to gadolinium. Injection of gadolinium-based contrast agents presents some risk for nephrogenic systemic fibrosis in patients with severe chronic kidney disease [48] and can accumulate in tissue [49] as well. DiaCEST agents have shown excellent potential for clinical translation based on a number of compounds possessing suitable biocompatibility and the implementation of CEST imaging sequences on General Electric, Siemens, and Philips Healthcare clinical scanners. Glucose and its derivatives are currently being evaluated for cancer imaging by a number of groups [50,51,52,53,54]; however, the small labile chemical shift (around 1.0 ppm) of these sugars leads to lower amplification factors in a region with a lot of other endogenous contrast limiting their detection sensitivity. In this study, we identified acyl hydrazones and acyl hydrazides with larger labile proton shifts (in the range of 2.8–5.0 ppm), which display strong CEST signals based on their moderate chemical exchange rates with water. Indeed, the N-acyl hydrazone scaffold has been identified as a promising motif in drug design and medicinal chemistry [45] based on the ease of synthesis and the clinical usage of several of these compounds for fighting bacterial infections, as hemostatic agents and to treat hyperthermia. Hydrazide analogs are used as antifungal, antibacterial, antiviral, and anticancer agents [55]. Furthermore, many also possess good water solubility and are highly biocompatible, enabling injections of high concentrations. Because of all these features, this scaffold has advantages over other NH- and NH_2_-based CEST agents, which are less biocompatible [24,25,38,56], which poses some challenges for the high doses required by the CEST technique.

We discovered ADH to be particularly well-suited for CEST imaging. Because of the large proton chemical shift (4.4 ppm), stronger saturation pulses can be utilized than those used on glucose. **ADH** is also very well tolerated with an oral LD_50_ of 5000 mg/kg for rats [57]. We observed no adverse reactions in our mice at the much lower imaging dosage of **ADH** (~290 mg/kg). In addition, the CEST contrast of **ADH** does not change between 6.1 > pH > 7.0, which makes it insensitive to pH over a wide range allowing for direct linkage between contrast changes to agent and concentration changes. The large chemical shift at 4.4 ppm and exchange rate in the slow-intermediate exchange regime make **ADH** a nice CEST contrast agent which should be readily detected on 3T scanners, and the in vivo imaging showed nice CEST contrast from **ADH** in tumors, indicating its potential in cancer imaging.

Furthermore, acyl hydrazones are a viable scaffold of CEST agents, which can possess larger chemical shifts (>6.0 ppm) than previously reported hydrazone compounds [27]. These have the additional advantage of simple synthesis from acyl hydrazides. These can also form gels at room temperature, as reported previously for two functionalized hyaluronic acid (HA) derivatives containing aldehyde (HA-aldehyde) or hydrazide (HA-hydrazide) [58,59]. However, in this article, they only mentioned the CEST effect from the -OH group at 1.0 ppm and not at 4.4 ppm or 6.4 ppm, which may be due to the low substitution of acyl hydrazide on HA to show visible CEST contrast at 4.4 or 6.4 ppm. Because of the simple synthesis and pH dependence of acyl hydrazone, this group could be applied to designing new pH sensors with larger chemical shifts with another labile proton incorporated to allow ratiometric measurements [38].

Our study, however, has several limitations. First, we only investigated a small subset of acyl hydrazides for their CEST properties. Strong electron withdrawing substitutions or amino next to the carbonyl group on acyl hydrazides could have a big impact on CEST contrast, for example, so we will need to test a larger library of hydrazones and hydrazides to fully understand how substitution will impact CEST contrast and identify the best probes. Second, in this study, we selected just one compound (**ADH**) to evaluate a mouse model of breast cancer and collected it’s in vivo CEST MRI over 34 min on a single slice instead of carrying out a detailed pharmacokinetic study. More types of acyl hydrazides should be evaluated in vivo as well. Though **ADH** was very well tolerated by mice during the imaging experiments, we will carefully test organ toxicity in the future. Third, acyl hydrazone showed great promise as a CEST agent with a large labile proton chemical shift (>6.0 ppm). Its biocompatibility and in vivo applications in cancer diagnosis were not investigated in this study as for pH imaging, an additional CEST active group with a different labile proton shift should be conjugated, which will be the subject of future investigations. Nevertheless, our work displays the promise of this scaffold for medical applications.

## 4. Materials and Methods

### 4.1. General Information and Chemistry

^1^H and ^13^C NMR spectra were recorded on a 300 MHz Bruker NMR spectrometer. Chemical shifts are in ppm, and coupling constants (J) are in Hertz (Hz). ^1^H NMR spectra were referenced to tetramethylsilane (d, 0.00 ppm) using CDCl_3_ as solvent. ^13^C NMR spectra were referenced to solvent carbons (77.16 ppm for CDCl_3_). The splitting patterns for ^1^H NMR spectra are denoted as follows: s (singlet), d (doublet), t (triplet), q (quartet), and m (multiplet). Mass spectra were recorded on an ESI mass spectrometer for compounds below 3000 Da.

All reagents were obtained from commercial suppliers and used without prior purification. Compounds **1**, **5**, **8**, **12**, **13,** and **14** were purchased from Bide Pharmatech (Shanghai, China). Compounds **2**, **6**, **7**, **15,** and **16** were obtained from Aladdin (Shanghai, China). Compounds **3**, **4**, **9**, **10,** and **11** were obtained from Macklin Biochemical Technology (Shanghai, China). All solvents were analytical or HPLC grade. Deionized water was used unless otherwise indicated.

### 4.2. Synthesis of Acyl Hydrazone **17**

**ADH** (348 mg, 2.0 mmol) was dissolved with 20 mL of anhydrous ethanol in a round bottle. **Aldehyde** (synthesized according to the literature [60], 1.72 g, 4.0 mmol) was dissolved in 20 mL anhydrous ethanol was added slowly. The mixture was refluxed for 6 h. The solvent was removed, and the crude product was purified by column chromatography on silica gel (CH_2_Cl_2_/MeOH = 20/1) to afford acyl hydrazone **17** as white waxy solid (1.53 g, 78% yield), ^1^H NMR (300 MHz, D_2_O) δ 7.26 (s, 2H), 7.21 (s, 2H), 6.95 (d, *J* = 9.0 Hz, 2H), 6.75 (d, *J* = 9.0 Hz, 2H), 3.96–4.00 (m, 8H), 3.70–3.73 (m, 8H), 3.58–3.60 (m, 8H), 3.49–3.55 (m, 16H), 3.42–3.46 (m, 8H), 2.19 (s, 4H), 1.56 (s, 4H). ^13^C NMR (75 MHz, D_2_O) δ 171.9, 149.9, 149.1, 147.7, 126.6, 123.3, 112.7, 110.2, 71.0, 69.9, 69.8, 69.6, 69.5, 68.9, 68.8, 67.9, 58.0, 33.8, 24.8. HRMS calcd for C_48_H_79_N_4_O_18_^+^ ([M + H]^+^) 999.5384, found 999.5369.

### 4.3. Phantom Preparation and In Vitro CEST MRI

Acyl hydrazides and acyl hydrazone were dissolved in 0.01M phosphate-buffered saline (PBS) at different concentrations and titrated to various pH values ranging from 5.7 to 7.5. The samples were kept at 37 °C during imaging. In vitro CEST MRI experiments were performed on a horizontal bore 9.4 T Bruker Biospec system (Bruker, Ettlingen, Germany). For phantoms, a transmit-receive volume coil with an inner diameter of 40 mm was used. CEST images were acquired using a continuous-wave RF saturation pulse of 3s followed by a RARE imaging sequence. CEST data were acquired using six different saturation power (B_1_) from 1.8, 2.4, 3.0, 3.6, 4.2, and 4.8 μΤ. One hundred two offsets between ±10 ppm were acquired with a 0.2 ppm stepsize to produce saturation images, plus one at +40 ppm. Other imaging sequence parameters: matrix size 64 × 64; slice thickness 1.5 mm; TR/TE 6000/4.6 ms. For B_0_ inhomogeneity calculation, water saturation shift referencing (WASSR) images were acquired using B_1_ = 0.5 μΤ with 31 offsets from −1.5 to +1.5 ppm with a 0.1 ppm stepsize.

### 4.4. In Vivo CEST MRI of 4T1 Tumor Mice Model

All animal experiments were performed under a protocol approved by the Jinan University Animal Care and Use Committee. Tumor xenografts of murine 4T1 breast cancer were established in three 4-week-old female Balb/c nude mice (Charles River Laboratories) through the subcutaneous injection of 2 × 10^6^ cells suspended in PBS above the rear right flank. Two to three weeks after cell injection, CEST MRI imaging was performed. In vivo CEST MRI data were collected on a horizontal bore 9.4 T Bruker Biospec system (Bruker, Ettlingen, Germany). Prior to imaging, the tail vein was cannulated, permitting intravenous injection in the MRI scanner. Mice were warmed, and their respiration rate was monitored throughout the duration of the imaging experiment. A T_2w_ image was obtained using a multi-slice RARE sequence. RARE factor = 8, TR/TE = 2 s/20 ms, field of view = 28 mm^2^ × 20 mm^2^, matrix size = 256 × 256, slice thickness 1.5 mm. For CEST data acquisition, 4 s long cw saturation pulses were applied with B_1_ = 3.6 μΤ using 31 offsets from −7.0 to +7.0 ppm with stepsize 0.2 ppm and one offset of +40 ppm for 72 offsets. WASSR images were collected for generating B_0_ maps using 31 offsets from −1.5 to +1.5 ppm with stepsize 0.1 ppm and B_1_ = 0.5 μT. After completion of the initial Z-spectrum, the mouse was injected with 100 μL of 500 mM (~435 mg/kg) **ADH**, and the line was flushed with 100 μL saline. No adverse events were noted in any of the mice receiving **ADH** injections. Post-contrast Z-spectra were acquired sequentially up to 34 min post-injection. The time taken for each set of 72 offsets was 8 min and 24 s, and the other sequence parameters were as follows: TR/TE 7000/4.6 ms; number of averages = 1; matrix size = 64 × 64; field of view 28 mm^2^ × 20 mm^2^.

### 4.5. CEST Data Analysis

All post-processing was performed using custom-written script implemented in MATLAB (MathWorks, Natick, Massachusetts). Saturation transfer (ST) contrast was calculated by applying magnetization transfer ratio asymmetry analysis with MTR_asym_ = (S_(−Δω)_ − S_(+Δω)_)/S_0_, where S_+Δω_ and S_−Δω_ were the MRI signals with RF irradiation at particular offsets +Δω and –Δω, respectively, and S_0_ is the signal acquired without RF saturation. Both S_+Δω_ and S_−Δω_ were corrected pixel-wise for B_0_ inhomogeneity using the WASSR method as described previously [61].

The exchange rates for the labile protons were estimated using the quantifying exchange using saturation power (QUESP) method as described previously [42]. In brief, the measured MTR_asym_ values at the maximum CEST offsets for the set of 6 saturation field strengths were fit numerically using the Bloch equations to estimate the exchange rates (k_ex_).

### 4.6. In Vitro Cellular Cytotoxicity Assay

4T1 cells were cultured in completed RPMI 1640 media (10% fetal bovine serum) and seeded into 96-well plates (100 μL/well) at a density of 1× 10^4^ cells per well for 24 h at 37 °C and in an atmosphere of 5% CO_2_. Then, the medium was removed and the cells were washed with PBS (×2). Afterward, the cells were exposed to different concentrations of **ADH** (0, 6.25, 12.5, 25, 50, 100 mM) in each well. The cells were grown for a further 24 h. Then a standard CCK-8 assay was carried out to evaluate the cell viability. Briefly, 10 μL of the CCK-8 solution was added to each well of the plate, and the cells were allowed to grow for another 3 h. The absorbance (optical density (OD)) was measured at 450 nm using a microplate reader (Epoch, BioTek Instruments Inc., Shoreline, WA, USA). The measured absorbance value (OD) of the wells was then used to calculate the cell viability via the formula
cell viability(%) = [(OD_experimental well_ − OD_blank_)/(OD_control well_ − OD_blank well_)] × 100%

## 5. Conclusions

In conclusion, we have identified a number of compounds from the acyl hydrazise–hydrazone scaffold, which produces strong CEST contrast from 2.8–6.4 ppm from water. We have also demonstrated that one compound from this scaffold (ADH) allows the detection of altered perfusion in a mouse model of breast cancer. Our study expands the catalog of compounds suited as CEST agents, which will accelerate the clinical translation of CEST MRI in the future.

## Data Availability

Data are contained within the article and Appendix A.

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
