# Peer review of "Acyl Hydrazides and Acyl Hydrazones as High-Performance Chemical Exchange Saturation Transfer MRI Contrast Agents"

_pharmaceuticals, 2023, doi:10.3390/ph16050639_

Round 1

Reviewer 1 Report

In the manuscript McMahon,  Shi,  Luo and collaborators describe the synthesis and the investigation of the properties as  chemical exchange saturation transfer (CEST) MRI of a wide range of organic compounds characterized by the presence of acyl hydrazide moieties with different substitutions including aromatic and aliphatic units. The detailed investigation allowed to find some target compounds with the labile proton chemical shifts from 2.8-5.0 ppm from water and exchange rates varied from ~680 to 2340 s-1 at pH 7.2. A symmetric water soluble compound bearing two acyl hydrazide units with up to 6.4 ppm shift from water and the excellent contrast properties was prepared and tested for cell toxicity as well as in vivo contrast experiments on mouse model breast cancer. Results are of high interest and potential applicability; full investigation is provided, and the topic is surely suitable for the journal. The paper is really well written, with clear description of the objectives of the work.

Overall I consider the manuscript a very good contribution that deserve publication with really minor changes in accordance with the below reported comments:

-In the abstract, I suggest changing “ We also prepare a derivative” with “ We also prepared a derivative”

-page 4, I think that  round brackets are unnecessary in “ADH (11), 7, and 14”

-In the Supporting information file correct integration of all resonances of the product in the 1H NMR spectrum. In the ESI spectra please report calculated and found m/z values.

Author Response

Response to Reviewer 1 Comments

Point 1: In the abstract, I suggest changing “ We also prepare a derivative” with “ We also prepared a derivative”.

Response 1: Thank you for your suggestion. We’ve corrected the sentence.

Point 2: page 4, I think that round brackets are unnecessary in “ADH (11), 7, and 14”.

Response 2: Thank you for your suggestion. We’ve corrected the sentence.

Point 3: In the Supporting information file correct integration of all resonances of the product in the 1H NMR spectrum. In the ESI spectra please report calculated and found m/z values.

Response 3: Thank you for your suggestion. We’ve added the calculated and found m/z values.

Reviewer 2 Report

The authors presented the paper "Acyl hydrazides and acyl hydrazones as high performance chemical exchange saturation transfer MRI contrast agents"

The author obtained an interesting result. I am positive about the possible publication of this work. However, I have some major comments regarding the paper.

1) I am a bit confused. Because of the Introdutcion, I see that the aim of the paper was to synthesize pH sensitive probe. However, the main results are not about it. ADH compound is not pH-sensitive. Please, re-change and change the prospects or enlarge the aim of the work in the Abstract, Introdution, and Conclusion according to the main results.

2) Figure 2. I see the differences between the pH of 5.7 and other values. How you can explain such results? Why no changes have been occurring at pH >6? Please, add a valuable comment in the paper text.

3) Lines 140-142. Can you explain to the readers how such conclusions have been done that ADH is an excellent contrast agent for depicting perfusion imaging?

4) Cell viability experiments show no cytotoxicity. Some of the hydrazones have systemic toxicity, mutagenic effect, and acute toxicity. Cell experiments may not provide it in a clear way. However, have you done histology experiments with your mouse after the imaging? Usually, liver toxicity may be occurring. Or maybe, some organs with the highest ADH concentration may suffer.

5) In vivo experiments. I see a low signal in the tumor. However, can you provide full mouse image with the injection place. Have you done time-dependent experiments? 

6) Figure 3 right. Are these results of the tumor? I see not good contrast of the tumor and surrounding tissues according to your figures on the left. How significant are these differences if you do not know that there is a tumor.

7) I see that you work using a 9.4 T imaging scanner. However, for the people 1.5 and 3 T are usually used. Is this in vivo imaging with your compound may works on such magnetic field?

8) I see limited results for acyl hydrazone 17. Have you done any in vivo experiments with this compound?

9) Discussion section is too poor. I don't see any desirable and reliable discussion about in vivo experiments, obtained spectra (Figures 2 and 4), and further applications of such compounds. Moreover, the main results was done using ADH and no valuable discussion was inserted about acyl hydrazone 17 and its further prospects and applications.

Author Response

Response to Reviewer 2 Comments

Point 1: I am a bit confused. Because of the Introduction, I see that the aim of the paper was to synthesize pH sensitive probe. However, the main results are not about it. ADH compound is not pH-sensitive. Please, re-change and change the prospects or enlarge the aim of the work in the Abstract, Introduction, and Conclusion according to the main results.

Response 1: Yes, this is correct. The aim of this paper is to expand the catalog of CEST contrast agents and investigate their application in cancer diagnosis. We have revised the Abstract, Introduction, and Conclusion according to the main results.

Point 2: Figure 2. I see the differences between the pH of 5.7 and other values. How you can explain such results? Why no changes have been occurring at pH >6? Please, add a valuable comment in the paper text.

Response 2: Thank you for your questions. The CEST contrast of ADH depends on the amide bond strength. The -NH2 on ADH can combine with H+ when the pH decreases to keep the amide bond stable, however, the stability of this amide bond on ADH is broken when the pH is ≤ 6.0 and this impacts the CEST contrast. We have added a comment now.

Point 3: Lines 140-142. Can you explain to the readers how such conclusions have been done that ADH is an excellent contrast agent for depicting perfusion imaging?

Response 3: ADH is well tolerated in cell culture. The CEST signal of ADH is not dependent on pH , so the signal is stable for a wide range of tissue pH values. The molecular weight of ADH is low and it has two functional -NH groups which can generate CEST signal and also it is highly water soluble. These features are all excelent for perfusion imaging. We have change the text to clarify this now.

Point 4: Cell viability experiments show no cytotoxicity. Some of the hydrazones have systemic toxicity, mutagenic effect, and acute toxicity. Cell experiments may not provide it in a clear way. However, have you done histology experiments with your mouse after the imaging? Usually, liver toxicity may be occurring. Or maybe, some organs with the highest ADH concentration may suffer.

Response 4: This is a good point, our study didn’t perform histology experiments with mouse after the imaging to validate if organs displayed damage. But according to the literature, ADH is very well tolerated at high does (Oral LD50 > 5g/kg for rat), a dose well above what we used to generate CEST contrast (~290 mg/kg) so we chose not to run these additional tests. We will test the toxicity of acyl hydrazides in a future more detailed study.

Point 5: In vivo experiments. I see a low signal in the tumor. However, can you provide full mouse image with the injection place. Have you done time-dependent experiments? 

Response 5: Unfortunately, we ran single slice images similar to other preclinical CEST studies and didn’t acquire full 3D datasets on these mice, this is difficult to do using a RARE sequence with a long saturation pulse based on acquisition times . We would also like to point out based on our coils, it would be quite difficult to acquire images all the way to the tail, as this is outside the coil. With that said, we did collect images at multiple time points as shown in Figure 3 and the contrast changes are similar in time course to what was observed for another imaging agent (glucose) in breast tumors (Chan et al MRM 2012 68: 1764-1773). In addition, the contrast changes are much larger than the signal to noise and are also greater than seen by Chan et al for glucose and a number of other CEST studies on tumors.

Point 6: Figure 3 right. Are these results of the tumor? I see not good contrast of the tumor and surrounding tissues according to your figures on the left. How significant are these differences if you do not know that there is a tumor.

Response 6: Thank you for your questions. The MTRasym results in Figure 3 right are inside the tumors. Actually, we can see CEST contrast building up in in ROIs in the tumor after intravenous injection of ADH. If we didn’t know there was a tumor, there is still clearly higher accumulation of contrast agents in this area, which indicates altered blood perfusion. The contrast is higher than seen for glucose in Chan et al MRM 2012 68: 1764-1773, which motivated translation of this agent to patients at 3T, so we believe this is also reasonable.

Point 7: I see that you work using a 9.4 T imaging scanner. However, for the people 1.5 and 3 T are usually used. Is this in vivo imaging with your compound may works on such magnetic field?

Response 7: Thank you for your questions. We didn’t run in vivo imaging with our compound at 1.5 or 3T magnetic field. However, we have experience evaluating 3 T CEST MRI performance based on the main properties that dictate this performance: labile proton chemical shift and chemical exchange rate. This compound has a suitable chemical shift and exchange rate as described in Yang et al Chem Euro J 2014 20: 15824-15832 and can work on 3T scanners. We have added a comment on this to the manuscript now.

Point 8: I see limited results for acyl hydrazone 17. Have you done any in vivo experiments with this compound?

Response 8: This is true, the paper was focused on investigating the CEST properties and applications of acyl hydrazides, so we didn’t run any in vivo experiments with acyl hydrazone 17. We wanted to identify the best imaging agent for perfusion and test that in vivo. Compound 17 has a strong pH dependence of the signal so it isn’t as good for perfusion imaging as ADH. In order for 17 to be used as a pH imaging agent, an additional CEST active group with a different labile proton shift should be conjugated which will be the subject of a follow up study.

Point 9: Discussion section is too poor. I don't see any desirable and reliable discussion about in vivo experiments, obtained spectra (Figures 2 and 4), and further applications of such compounds. Moreover, the main results was done using ADH and no valuable discussion was inserted about acyl hydrazone 17 and its further prospects and applications.

Response 9: Thank you for your suggestions. We have rewritten the discussion section about in vivo experiments, Figures 2 and 4, and added further commentary on acyl hydrazone 17 and its further prospects as well.

Round 2

Reviewer 2 Report

Thank you for the revised paper.